# High spin axion insulator

Shuai Li [1,2], Ming Gong [3], Yu-Hang Li [4] ✉, Hua Jiang [2,5,6] ✉ & X. C. Xie [3,5,6,7]

Axion insulators possess a quantized axion field $\theta = \pi$ protected by combined lattice and time-reversal symmetry, holding great potential for device applications in layertronics and quantum computing. Here, we propose a high-spin axion insulator (HSAI) defined in large spin-$s$ representation, which maintains the same inherent symmetry but possesses a notable axion field $\theta = (s + 1/2)^2 \pi$. Such distinct axion field is confirmed independently by the direct calculation of the axion term using hybrid Wannier functions, layer-resolved Chern numbers, as well as the topological magneto-electric effect. We show that the guaranteed gapless quasi-particle excitation is absent at the boundary of the HSAI despite its integer surface Chern number, hinting an unusual quantum anomaly violating the conventional bulk-boundary correspondence. Furthermore, we ascertain that the axion field $\theta$ can be precisely tuned through an external magnetic field, enabling the manipulation of bonded transport properties. The HSAI proposed here can be experimentally verified in ultra-cold atoms by the quantized non-reciprocal conductance or topological magnetoelectric response. Our work enriches the understanding of axion insulators in condensed matter physics, paving the way for future device applications.

Symmetry plays an essential role in understanding the behavior of condensed materials[1-4]. For example, in the presence of time-reversal symmetry, three dimensional insulator typically falls into two categories: one is trivial insulator while the other is topological insulator[5,6]. These divergent categories can be well described within the framework of the Chern-Simons theory, where the Lagrangian incorporates an additional symmetry allowed term $\mathcal{L}_\theta = \int dt d\mathbf{r}^3 \alpha\theta/(4\pi^2) \mathbf{E} \cdot \mathbf{B}$ with $\mathbf{E}$ and $\mathbf{B}$ the conventional electric and magnetic fields, $\alpha$ the fine structure constant, and $\theta$ the gauge dependent axion term[7]. Because of the $2\pi$ periodicity under a gauge transformation[8], the axion term here is well defined within the region ($[0, 2\pi)$). Besides, as the quantity $\mathbf{E} \cdot \mathbf{B}$ flips sign under time-reversal ($\mathcal{T}$) operation, the axion field manifests only two distinct values, that is, $\theta = 0$ for normal insulator and $\theta = \pi$ for topological insulator[7]. Furthermore, the non-vanishing axion term in the Lagrangian would introduce additional magneto-electric responses to the Maxwell equations and in turn, results in a distinctive topological magneto-electric effect[9-11], furnishing a hallmark to the quantized axion field.

In addition to the time-reversal ($\mathcal{T}$) symmetry, the quantized axion field $\theta = \pi$ can also be protected by combined lattice and time-reversal symmetry (for example $\mathcal{S} = \mathcal{T}\tau_{1/2}$ with $\tau_{1/2}$ the half translation operator)[12], as the quantity $\mathbf{E} \cdot \mathbf{B}$ undergoes the same sign reversal. This kind of materials, termed axion insulator[13-24], holds significant potential in layertronics[25-29] and quantum computing[30,31]. $MnBi_2Te_4$ and its family have recently been proposed as axion insulators in the antiferromagnetic state[13,32-36], which finds a concise description with an effective Hamiltonian defined on the orbital and spin-1/2 spaces[33]. Because the symmetry transformations of high spin representations and spin-1/2 are identical (see Supplementary Table 1), in this work, we generalize this model to other spin species and thus propose a high spin axion insulator (HSAI) preserving the same symmetry. We find that the HSAI with spin-$s$ possesses a notable axion field $\theta_{HSAI} = (s + 1/2)^2\pi$. It carries a multiple quantized helical hinge current (QHHC) that is robust against disorders even in the absence of the topologically protected gapless excitations, which contradicts the integer surface

---

[1]School of Physical Science and Technology, Soochow University, Suzhou 215006, China. [2]Institute for Advanced Study, Soochow University, Suzhou 215006, China. [3]International Center for Quantum Materials, School of Physics, Peking University, Beijing 100871, China. [4]School of Physics, Nankai University, Tianjin 300071, China. [5]Institute for Nanoelectronic Devices and Quantum Computing, Fudan University, Shanghai 200433, China. [6]Interdisciplinary Center for Theoretical Physics and Information Sciences (ICTPIS), Fudan University, Shanghai 200433, China. [7]Hefei National Laboratory, Hefei 230088, China. ✉e-mail: liyuhang@nankai.edu.cn; jianghuaphy@fudan.edu.cn

Chern number. Consequently, HSAI exhibits an unusual quantum anomaly that violates the conventional bulk-boundary correspondence. In contrast to the case of spin-1/2 axion insulator, the direct calculation of the axion term shows that the large axion field in high-spin case originates mostly from localized surface Wannier functions while, in the bulk, the axion field is either 0 or $\pi$. Strikingly, we show that the axion field in HSAI can be tuned precisely by manipulating the magnetic configuration through an external magnetic field, providing a pioneering tuning knob to control the QHHC and the quantized magneto-electric response. Possible experimental realization in ultra-cold atoms is also discussed.

## Results

### Effective model for the HSAI

Recalling the effective four-band Hamiltonian for the spin-1/2 axion insulator[33], we consider a generic model defined on the high spin space which can be expressed as

$$\mathcal{H} = \sum_{i=0}^{3} d_i \Gamma_i + \Delta \mathbf{m}_s \cdot \mathbf{s} \otimes \tau_0. \quad (1)$$

Here, $d_{0,1,2,3} = [m_0 - Bk^2, \; Ak_x, \; Ak_y, \; Ak_z]$ with $A, B, m_0$ the system parameters. $\Gamma_0 = s_0 \otimes \tau_3$ and $\Gamma_{i=1,2,3} = s_i \otimes \tau_1$ where $s_i$ and $\tau_i$ are matrices defined on the high spin space and $2 \times 2$ orbital space, respectively. The momentum $\mathbf{k} = (k_x, k_y, k_z)$ is defined on a cubic lattice with the lattice constant $a_0$ inside the first Brillouin zone. This model Hamiltonian is given directly from the spin-1/2 axion insulator. A construction from symmetry perspective is provided in Supplementary Note 2. It is evident that the first term in Eq. (1) describes a high-spin topological insulator, which preserves both time-reversal ($\mathcal{T}$) and parity ($\mathcal{P}$) symmetries. The second term describes the exchange interaction between topological electrons and normalized magnetic spins $\mathbf{m}_s = (m_s^x, m_s^y, m_s^z)$ with $\Delta$ the exchange strength, resembling that

in MnBi$_2$Te$_4$, hence explicitly breaks the time-reversal symmetry while preserves the $\mathcal{S}$ symmetry in the infinite size limit along $z$-direction. We consider the antiferromagnetic phase of an even-layer slab involving only the antiparallel (or canted) spins in the top and bottom layers as illustrated in Fig. 1a, which restores the combined parity and time-reversal ($\mathcal{PT}$) symmetry. Unless otherwise specified, we adopt the typical model parameters as follows: $A = m_0 = 1$, $B = \Delta = 0.6$, $a_0 = 1$.

Figure 1 (b) displays the two dimensional energy spectra of the spin-3/2 HSAI in the absence (blue dashed lines) and presence (red solid lines) of the magnetic exchange term. In the former case, the time-reversal symmetry is present, where the energy spectrum is gapped in the bulk but has two conducting surface states on each side (blue dashed lines). These two surface states can be accurately fitted by a massless Dirac band $E_1 \sim k$ and a cubic band $E_2 \sim k^3$ (inset in Fig. 1b). Turning on the exchange term in the latter case opens a band gap as indicated by the red solid lines in Fig. 1b. In both cases, the energy spectra are doubly degenerated because of the inherent ($\mathcal{T}$ or $\mathcal{PT}$) symmetry.

### Layer-resolved Chern number, quantized helical hinge current and quantum anomaly

To explore the topological properties of the HSAI, we calculate the layer-resolved Chern number $C_z$ along $\hat{z}$-direction[32,37] along with the cumulative Chern number $\tilde{C}_z = \sum_{-L_z/2}^{z} C_z$. Given the Chern number $C = (s+1/2)^2$ in odd-layer case[38], the system is a high Chern number insulator as shown in Supplementary Note 6. In the even layer system, the opposite layer-resolved Chern numbers are overall confined anti-symmetrically inside few surface (top and bottom) layers as shown in Fig. 1c, resulting in a vanishing total Chern number $C = 0$. Nevertheless, the surface Chern number on one side turns out to be well quantized [$C_{surf}^{top(bot)} = \mp 2$] when $s = 3/2$ as long as the Fermi level lies inside the energy gap (Fig. 1d). Because the layer-resolved Chern number is

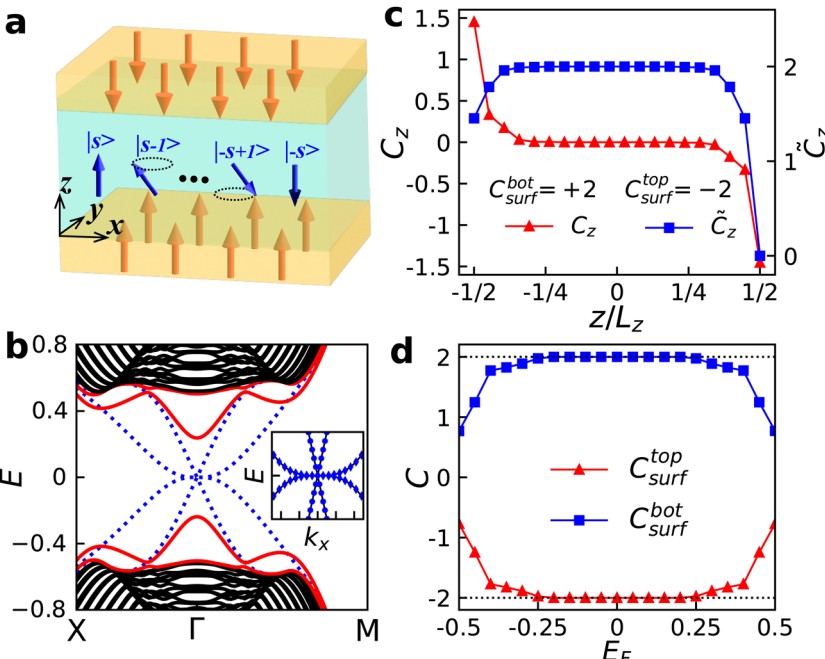

**Fig. 1 | Model of the HSAI. a** Schematic for the HSAI defined on the $|s, m_z\rangle$ space. The blue arrows represent the electron spin with different magnetic quantum number $m_z$, which takes values ranging from $-s$ to $s$ individually. **b** Energy spectra of the spin-3/2 HSAI along M → Γ → R path on a slab of thickness $L_z$ with (red solid lines) and without (blue dashed lines) the magnetic exchange interaction. Here, the black lines refer to bulk bands. Inset: Energy dispersion for the spin-3/2 HSAI in the absence of magnetic exchange term near the charge neutral point (solid lines) and

the fitting data (markers). **c** Layer-resolved Chern number $C_z$ and the cumulative Chern number $\tilde{C}_z = \sum_{-L_z/2}^{z} C_z$ versus the layer index $z$ for the spin-3/2 HSAI. The surface Chern numbers $C_{surf}^{top(bot)}$ that summarize the layer-resolved Chern number on the upper (lower) half side is $-2$ ($+2$). **d** Surface Chern number as a function of the Fermi energy $E_F$ for the spin-3/2 HSAI. Here, the thickness of the HSAI slab is $L_z = 20$.

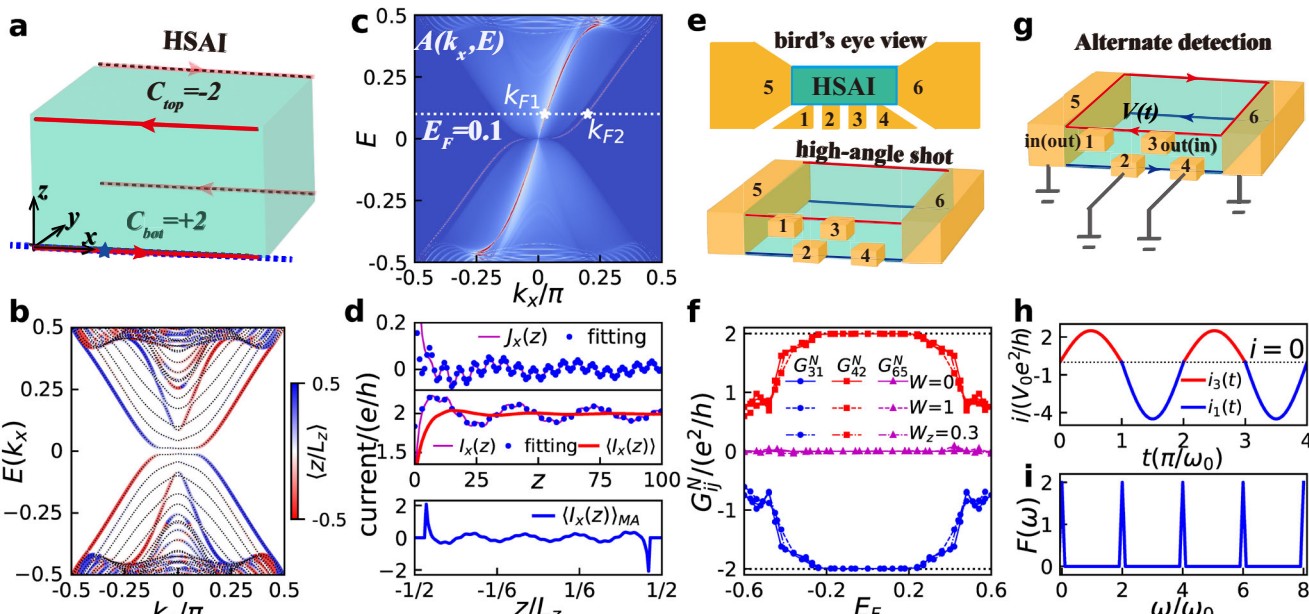

**Fig. 2 | Transport properties of the spin-3/2 HSAI. a** Schematic current flow in a HSAI. The red arrows denote the quantized helical hinge currents. **b** Energy spectrum and the average position $\langle z/L_z \rangle$ on the front surface for a HSAI nanowire with $L_y = 30$, $L_z = 16$. **c** Spectrum density $A(k_x, E)$ for the front lower hinge as labeled by the blue line in (**a**) on the $k_x - E$ plane. Here, the system size is $L_y = 30$, $L_z = \infty/2$. The white dashed line represents the Fermi energy $E_F = 0.1$. The white stars that mark the intersects between the Fermi energy and the spectrum are the Fermi momenta $k_{F1}$ and $k_{F2}$. **d** Top and middle panels are the current density $J_x(z)$, current flux $I_x(z)$ and its $z$-averaged flux $\langle I_x(z)\rangle$ versus the layer index $z$ for a semi-infinite system with size $L_y = 30$, $L_z = \infty/2$. The blue dots are the fitting data. Bottom panel shows the distribution of the moving averaged current $\langle I_x(z)\rangle_{MA}$ on the front

surface with system size $L_y = 30$, $L_z = 150$. **e** Bird's eye view (top panel) and high-angle shot (bottom panel) for the six terminal device. **f** Ensemble-averaged non-reciprocal conductances versus the Fermi energy in the clean limit (W=0), with non-magnetic Anderson disorders of strength $W = 1$ and with magnetic Anderson disorders of strength $W_z = 0.3$. Here, the system size is $L_x = 31$, $L_y = 20$, $L_z = 21$, and the size of transverse terminals is $10 \times 10$. **g** Experimental setup to detect the non-reciprocal conductance. In this setup, terminals 2, 4, 5, and 6 are grounded. The voltage is applied alternatively to terminal 1 or terminal 3. **h** Corresponding temporal dependent current output with parameters $G_{13} = 4.5e^2/h$, $G_{31} = 2.5e^2/h$. **i** $F(\omega)$ as a function of the frequency $\omega$.

related to the axion field through the relation $\theta_{HSAI} = (C_{surf}^{bot} - C_{surf}^{top})\pi$[39]. The oppositely quantized surface Chern numbers in spin-3/2 HSAI thereby assure a quadruple axion field $\theta_{HSAI} = 4\pi$. Moreover, since the Chern number difference between neighboring top (bottom) and side surfaces is an integer, HSAI supports a possible hinge state that is absent in spin-1/2 axion insulator[26,40], which allows a subsequent QHHC owing to opposite chiralities on different surfaces. Nonetheless, we find that this QHHC survives counterintuitively without the existence of any hinge state.

To clarify this, let us first examine the average position $\langle z/L_z \rangle$ on the front surface of the slab at $y/L_y = -1/2$. The results shown in Fig. 2b reveals two branches (red lines in Fig. 2b) concentrated unexpectedly around the bottom hinge ($z/L_z = -1/2$) and propagating rightwards because of the positive group velocity. By contrast, we observe two other branches concentrate oppositely around the top hinge ($z/L_z = 1/2$), which, simultaneously, propagate leftwards as depicted by the blue lines. Note that only the results for the front surface (at $y/L_y = -1/2$) are presented here. In the presence of $\mathcal{PT}$ symmetry, the energy spectrum in Fig. 2b is doubly degenerated as stated above. There are four additional branches existing on the other surface at $y/L_y = 1/2$. Because the wavefunctions on the diagonal hinges are connected by this $\mathcal{PT}$ symmetry, they must propagate along the same direction, supporting a helical hinge current.

We then turn to the spectrum density $A(k_x, E)$ on a semi-infinite slab[41–43], where the system extends infinitely along $+\hat{z}$-direction but remains unchanged along the lateral directions. $A(k_x, E)$ on the front lower hinge illustrated by the blue dashed line in Fig. 2a are plotted in Fig. 2c. It shows that $A(k_x, E)$ originates mostly from the right-moving energy bands, agreeing remarkably well with the average position in Fig. 2b. This spectrum density can be verified

experimentally by using the nano angle-resolved photoemission spectroscopy and microscopy[44–46]. Besides, the high spectrum density on the hinge indicates the presence of a hinge current, the current density of which can be quantitatively determined by[26,43]

$$j_x(E_F, \mathbf{r}) = -\frac{e}{h\pi}\int_{-\pi}^{\pi} dk_x \text{Im}\{\text{Tr}[\frac{\partial H_{HSAI}(k_x, \mathbf{r})}{\partial k_x} G^r(E_F; k_x, \mathbf{r})]\}, \quad (2)$$

where $E_F$ is the Fermi energy labeled by the white line in Fig. 2c, $\mathbf{r} = (y, z)$, $H_{HSAI}(k_x, \mathbf{r})$ is the Hamiltonian for the HSAI and $G^r(E_F; k_x, \mathbf{r})$ is the retarded Green's function.

The upper panel in Fig. 2d presents the hinge current density $J_x(z) = \sum_{y=-L_y/2}^{0} j_x(\mathbf{r})$ as a function of the layer index $z$. We see that $J_x(z)$ is verily confined on the hinge, in agreement with $\langle z/L_z \rangle$ and $A(k_x, E)$. Strikingly, this hinge current decays oscillatively into the side surface, exhibiting a beating mode (magenta line) in sharp contrast to that in spin-1/2 axion insulator[26]. This peculiar behavior can be quantitatively fitted by the superposition of two power-law decaying edge currents $J_x^1(z) = \frac{a_1}{\sqrt{z}}\cos(2k_{F1}z + \phi_1)$ and $J_x^2(z) = \frac{a_2}{\sqrt{z}}\cos(2k_{F2}z + \phi_2)$[47], where $k_{F1}$ and $k_{F2}$ are the Fermi momenta for the two distinct modes marked by the white stars in Fig. 2c, while $a_{1(2)}$ and $\phi_{1(2)}$ are fitting parameters. The integral of the current density over the layer index provides the current flux $I_x(z) = \int_0^z d\tilde{z}J_x(\tilde{z})$ (middle panel in Fig. 2d), which oscillates around $2e/h$ and coincides perfectly with the fitting data. Additionally, the $z$-averaged current $\langle I_x(z)\rangle = \int_0^z d\tilde{z}I_x(\tilde{z})/z$ (red line) quantizes to $2e/h$ only a few layers away from the hinge. Imposing a finite thickness along $\hat{z}$-direction enables us to calculate the moving average current $\langle I_x(z)\rangle_{MA} = \int_{z-7}^{z+7} d\tilde{z}J_x(\tilde{z})$. The result displayed in the bottom panel demonstrates a helical hinge current quantized precisely to $\pm 2e/h$. Although the HSAI supports a QHHC identical to its integer surface

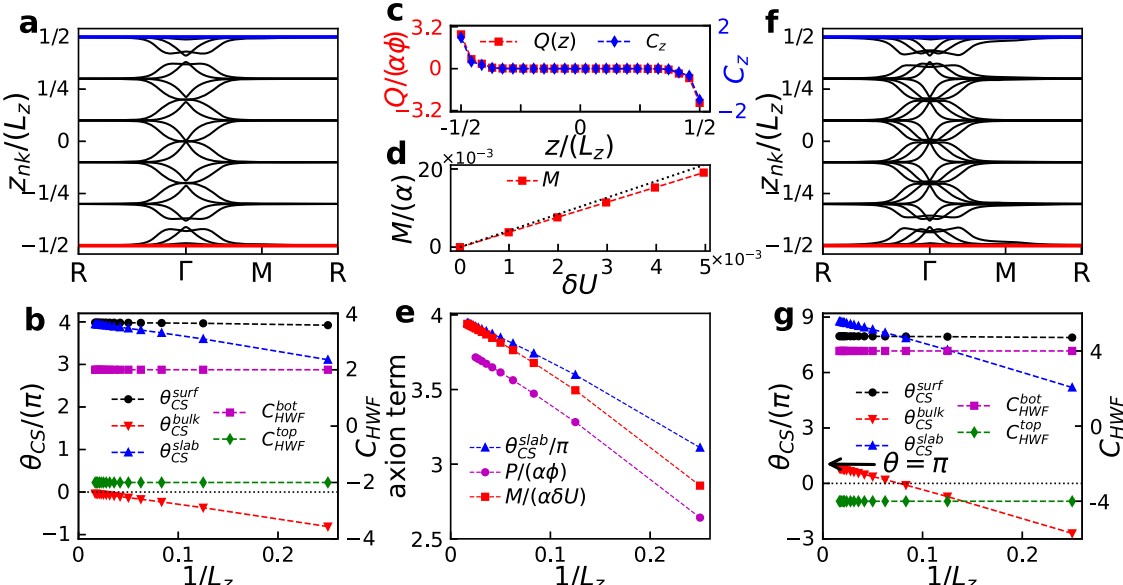

**Fig. 3 | Axion term and topological magneto-electric effect. a** and (**f**) Hybrid Wannier charge centers $z_{n\mathbf{k}}$ along R → Γ → M → R loop inside the first Brillouin zone for a six-layer HSAI slab with spin-3/2 (**a**) and spin-5/2 (**f**), respectively. **b** and **g** are corresponding axion terms and the surface Chern numbers versus the inverse layer thickness obtained by using the HWFs. **c** Magnetic field induced charge distribution along $\hat{z}$-direction and the layer-resolved Chern number for a spin-3/2 HSAI with $L_z = 24$. Here, the charge polarization is obtained on a HSAI slab with open boundary condition along $\hat{y}$-direction ($L_y = 40$) but periodic boundary condition along $\hat{x}$-direction. The magnetic flux inside one unit cell is $\phi_0 = Ba_0^2 = 0.01h/e$. **d** Electric field induced orbital magnetization for a spin-3/2 HSAI with $L_z = 20$. The black dashed line shows the ideal case (IC) with an exact axion term $\theta = 4\pi$. **e** Size scaling of the axion term $\theta_{CS}^{slab}/\pi$, polarization coefficient $P/(\alpha\phi)$, and magnetization coefficient $M/(\alpha\delta U)$ at $\delta U = 0.001$. We have checked that the slight deviation of $P/(\alpha\phi)$ originates from the finite size effect.

Chern number, the topologically protected gapless excitations in lower dimension are completely absent as plotted in Fig. 2b. It worths note that the energy gap in Fig. 2b may be induced by the finite size effect. Our analysis in Supplementary Note 3 shows that the size dependence of the energy gap comes from the bulk bands, which demonstrates that this energy gap originates from the magnetic exchange interaction. Consequently, the conventional bulk-boundary correspondence that an integer Chern number must hold chiral edge states fails in HSAI, establishing an unusual quantum anomaly.

**Non-reciprocal conductance**

Owing to the chirality bonded to the quantized surface Chern number, this QHHC can be unveiled by the non-reciprocal conductance $G_{ij}^N = G_{ij} - G_{ji}$ in a six-terminal device sketched in Fig. 2e, where $G_{ij}$ is the differential conductance[26,48]. In this device, two longitudinal leads (terminals 5 and 6) are intimately connected to the two ends of the sample while four transverse leads (terminals 1, 2, 3, and 4) are attached to different hinges on the front surface. Figure 2f shows three representative non-reciprocal conductances versus the Fermi energy $E_F$ in the clean limit (solid lines), with non-magnetic Anderson disorder (dashed lines) and with magnetic Anderson disorder (dashed dotted lines). In general, $G_{65}^N = 0$, $G_{31}^N = -2e^2/h$ and $G_{42}^N = 2e^2/h$ when the Fermi level lies inside the band gap for all three cases, consistent with the current distribution in Fig. 2d as well as the layer-resolved Chern numbers in Fig. 1d. This verifies that the QHHC is topological protected as the quantized non-reciprocal conductance is immune to both non-magnetic and magnetic Anderson disorders that even breaks the global $\mathcal{PT}$ symmetry[5,6].

Since multiple frequency ac current is robust against ambient perturbation, to detect this QHHC, we employ an alternate detection in which terminals 2, 4, 5, and 6 are grounded whereas a harmonic voltage $V(t) = V_0 \sin(\omega_0 t)$ with a periodicity $T = 2\pi/\omega_0$ is applied alternatively to terminal 1 or 3 as illustrated in Fig. 2g. During the first (second) half period, a positive (negative) voltage is applied to terminal 1 (3) as an input while the current flows $i_3(t)$ [$i_1(t)$] from terminal 3

(1) is detected as an output. Their combination gives rise to an asymmetric net current $i(t) = i_1(t) + i_3(t)$ as shown in Fig. 2h. Performing a Fourier transform converts $i(t)$ into $I(\omega)$. The non-reciprocal conductance can then be determined from the equation $F(\omega) = |I(\omega)(\omega^2 - \omega_0^2)|/(2N\omega_0 V_0)$ with $N$ the number of periodicity (see Supplementary Note 4 for details). The result displayed in Fig. 2i shows that $F(\omega) = G_{13}^N$ when $\omega = 2\omega_0$. Thus, non-reciprocal conductance measurements offer a reliable experimental method to visualize the QHHC in HSAI.

**Axion term**

The HSAI can alternatively be characterized by the axion term[7], which can be evaluated directly from the hybrid Wannier functions (HWFs) constructed in terms of the Bloch wavefunctions[49]. In this scenario, the axion term on a slab is defined as[49]

$$\theta_{CS}^{slab} = -\frac{1}{L_z} \int d^2\mathbf{k} \sum_n \left[ z_{n\mathbf{k}} \tilde{\Omega}_{\mathbf{k}nn}^{xy} \right], \tag{3}$$

where $z_{n\mathbf{k}}$ is the hybrid Wannier charge center along $\hat{z}$-direction and $\tilde{\Omega}_{\mathbf{k}nn}^{xy}$ is corresponding non-Abelian Berry curvature.

Figure 3 (a) shows $z_{n\mathbf{k}}$ in the first Brillouin zone for a six-layer HSAI with spin-3/2. These $z_{n\mathbf{k}}$ consist of two different types, those localized on the top and bottom surfaces as emphasized by the red and blue lines and those extending into the bulk denoted by black lines. Those surface Wannier bands will disappear under a periodic boundary condition when connecting the top and bottom surfaces. The total axion term of the slab can subsequently be divided into two parts $\theta_{CS}^{slab} = \theta_{CS}^{bulk} + \theta_{CS}^{surf}$ with $\theta_{CS}^{bulk}$ and $\theta_{CS}^{surf}$ the axion terms corresponding to the surface and bulk HWFs. The bulk axion term $\theta_{CS}^{bulk}$ is identical to that obtained analytically from the Chern-Simons three form in the infinite size limit[49]. In Fig. 3b, we show $\theta_{CS}^{bulk}$ (red upside down triangle), $\theta_{CS}^{surf}$ (black circle) together with $\theta_{CS}^{slab}$ (blue triangle) versus the inverse thickness $1/L_z$. There are three distinctive features in this figure. First, the total axion term shows an obvious tendency quantized to $\theta_{CS}^{slab} = 4\pi$ when the system size approaches infinity ($1/L_z \to 0$), which confirms the

quadruple axion term in two dimensional HSAI slab. Second, the axion term originates completely from the surface HWFs although the bulk HWFs also result in a small value that decreases $\theta_{CS}^{bulk}$ at finite size. Third, the axion term $\theta_{CS}^{surf}$ obeys the relation $\theta_{CS}^{surf} = (C_{HWF}^{bot} - C_{HWF}^{top})\pi$ in the infinite size limit, where $C_{HWF}^{top(bot)}$ is the top (bottom) surface Chern number obtained from the surface HWFs as indicated by cyan diamond (magenta square). These peculiar results reaffirm the unusual quantum anomaly and also the quadruple axion term $\theta_{HSAI} = 4\pi$ in HSAI with spin-3/2.

## Topological magneto-electric effect

Such quadruple axion term implies a unique topological magneto-electric effect[13,32]. When applying a magnetic field $B_z$ to the HSAI along $\hat{z}$-direction, the Hamiltonian in Eq. (1) acquires a Peierls phase[32], which redistributes the electron charge $Q(z)$ in accordance to the confined layer-resolved Chern number $C_z$ as shown in Fig. 3c. The ensuing charge polarization $P = \sum_{z=-L_z/2}^{L_z/2} zQ(z)/L_z$ almost quantizes to $P \approx 4\alpha\phi$, where $\alpha$ is the fine structure constant and $\phi$ is the total magnetic flux penetrating the HSAI slab. In comparison, a quantized orbital magnetization can emerge under an external electric field $E_z$ when incurring a potential drop $\delta U = eE_z L_z$ in the HSAI Hamiltonian[50]. The red square in Fig. 3(d) shows the orbital magnetization $M$ as a function of $\delta U$, which agrees quantitatively

well with the ideal case benchmarked by the black line. These two results independently certify the quadruple axion term $\theta_{HSAI}$ in spin-3/2 HSAI. The slight deviation from the exactly quadruple value originates from the finite size effect, which is further revealed by the size scalings of the axion term $\theta_{CS}^{slab}/\pi$, polarization coefficient $P/(\alpha\phi)$, and magnetization coefficient $M/(\alpha\delta U)$ against the inverse layer thickness $1/L_z$ in Fig. 3e. We also evaluate the axion term and the surface Chern numbers for spin-5/2 HSAI in terms of the HWFs (Fig. 3f). The results displayed in Fig. 3g demonstrate that spin-5/2 HSAI possesses a surface Chern number $C_{HWF}^{top(bot)} = \mp 4$, a total axion term $\theta_{CS}^{slab} = 9\pi$, a surface axion term $\theta_{CS}^{surf} = 8\pi$ and a bulk axion term $\theta_{CS}^{bulk} = \pi$. Systematic results for the spin-5/2 HSAI are provided in Supplementary Note 5. The topological properties for HSAI with different spin species are summarized in Table 1, giving a distinct axion field $\theta = (s+1/2)^2\pi$ and $C_{surf}^{top/bot} = \mp(1/2 + 3/2 + \cdots + s)$.

## Tunable topological phase transition

In the presence of an in-plane magnetic field, the antiparallel magnetic moments in the top and bottom layers become canted with the canting angle $\gamma$ proportional to the magnetic field strength as illustrated in Fig. 4a. In this case, the quantized axion field in the infinite size limit is protected by $m_x\mathcal{P}$ symmetry where $m_x$ is the mirror plane normal to $x$-direction. In Fig. 4b, we compare two dimensional band gaps as functions of $\gamma$ for spin-1/2 axion insulator and spin-3/2 HSAI. Because the exchange gap is determined by the perpendicular magnetization $M_z$, the band gap for spin-1/2 axion insulator decreases monotonically as $\gamma$ is enlarged and finally becomes zero when $\gamma = \pi/2$. On the contrary, the band for spin-3/2 HSAI exhibits a gap close at $\gamma = \pi/4$ as shown in Fig. 4c, suggesting a possible topological phase transition. Indeed, Fig. 4e shows that the surface Chern number obtained using both the Bloch wavefunctions and the HWFs(Fig. 4d) changes from $+2$ ($-2$) to $+1$ ($-1$) when $\gamma = \pi/4$. At this point, the HWFs are connected at the $\Gamma$ point (Fig. 4d), therefore the Berry curvature and the surface Chern number can transfer from one side to the other[51], leading to an axionic

**Table 1 | Axion terms and surface Chern numbers for axion insulators with different spins**

| spin-s | $\theta_{CS}^{slab}$ | $\theta_{CS}^{surf}$ | $\theta_{CS}^{bulk}$ | $C_{HWF}^{top}$ | $C_{HWF}^{bot}$ | $C_{surf}^{top}$ | $C_{surf}^{bot}$ |
|---|---|---|---|---|---|---|---|
| 1/2 | $\pi$ | 0 | $\pi$ | 0 | 0 | -1/2 | 1/2 |
| 3/2 | $4\pi$ | $4\pi$ | 0 | -2 | 2 | -2 | 2 |
| 5/2 | $9\pi$ | $8\pi$ | $\pi$ | -4 | 4 | -9/2 | 9/2 |
| 7/2 | $16\pi$ | $16\pi$ | 0 | -8 | 8 | -8 | 8 |
| ⋮ | | | | | | | |

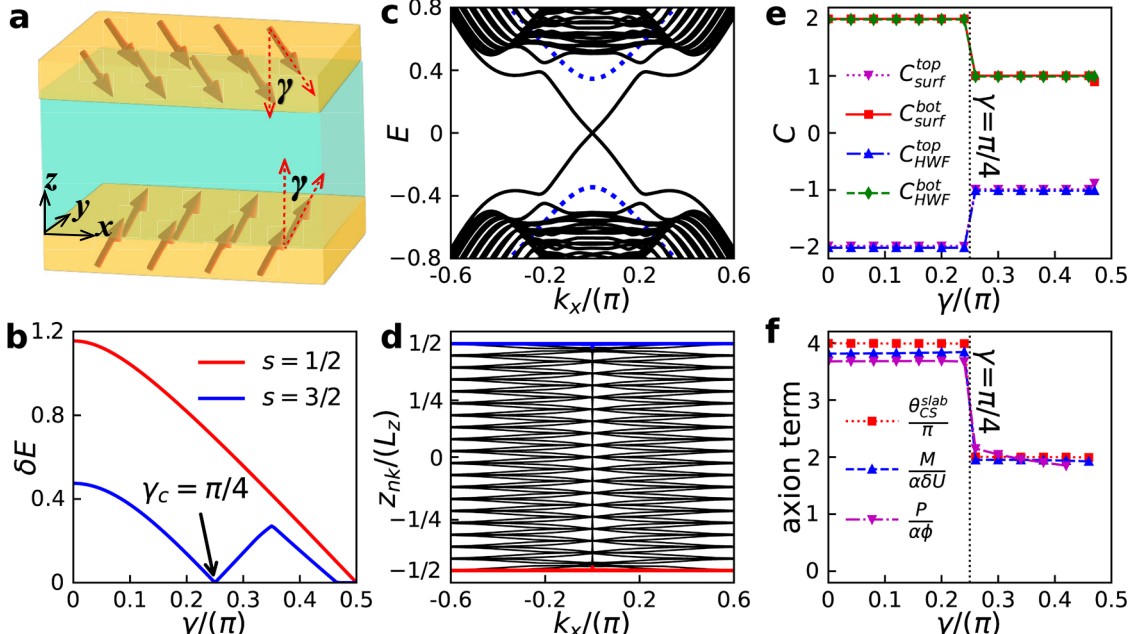

**Fig. 4 | Phase transition in spin-3/2 HSAI. a** Canted HSAI under an in-plane magnetic field. $\gamma$ is the canting angle between the magnetic vector and $\hat{z}$-axis (polar angle). **b** Energy gaps versus $\gamma$ for spin-3/2 HSAI and for spin-1/2 axion insulator, respectively. **c** Energy spectra for the HSAI with spin-3/2 (black solid lines) and spin-1/2 (blue dashed lines) at $\gamma = \pi/4$. **d** Hybrid Wannier charge center $z_{n\mathbf{k}}$ as a function of $k_x$ for a HSAI slab at $\gamma = \pi/4$. **e** Surface Chern numbers obtained from the effective Hamiltonian in Eq. (1) and the HWFs versus $\gamma$. **f** Axion term $\theta_{CS}^{slab}/\pi$, polarization coefficient $P/(\alpha\phi)$ ($\phi_0 = 0.01h/e$) and magnetization coefficient $M/(\alpha\delta U)$ ($\delta U = 0.001$) versus $\gamma$. The thickness of the HSAI is $L_z = 20$.

phase transition. Such topological phase transition is further affirmed by the axion field, the polarization and magnetization coefficients shown in Fig. 4f.

## Discussion

The device application of axion insulators requires the fine-control of the transport signals such as the magneto-electric response or the QHHC, which are identical to the axion field. In spin-1/2 axion insulators, the axion term cannot be tuned without disrupting the existing $\mathcal{S}$ symmetry or refabricating the setup[52]. Nevertheless, because different surface bands shown in Fig. 1b can be coupled via the in-plane exchange interaction ($M_x s_x \otimes \tau_0$), an apparent topological phase transition between axion insulators with different axion fields occurs in the HSAI. Consequently, the axion term $\theta_{HSAI}$ (in unit of $\pi$), hence the QHHC $G_{ij}^N$ (in unit of $e^2/2h$) and the magneto-electric effect $P/(\alpha B)$ [$M/(\alpha E)$] in HSAI can be precisely adjusted from 4 to 2 via the application of an external magnetic field. Thus, our work opens up an exciting possibility for the groundbreaking advancement in the practical application of axion insulators.

In conclusion, we have proposed a HSAI defined on the high spin space and shown that this HSAI possesses a multiple axion field protected by the combined lattice and time-reversal symmetry. Notably, the axion term in the bulk of a HSAI still quantizes to $\theta = 0$ or $\theta = \pi$ while the surface of HSAI possesses a large axion term and a consistent integer Chern number, which can be tuned by manipulating the magnetic configuration through an external magnetic field. These results extend the scope of recently discovered axion insulator in magnetic topological materials. In ultra-cold fermions on a honeycomb lattice, the exchange gap can be introduced by complex next-nearest-neighbour tunneling terms through circular modulation of the lattice position[53]. We thus propose that our theory can be tested in high spin ultra-cold fermions on a stacked honeycomb lattice, where the non-reciprocal conductance can be detected by the orthogonal drifts analogous to a Hall current under a constant force to the atoms[53,54].

## Methods

### Caltulations of the layer-resolved Chern number, magnetization, and polarization

In a HSAI slab with periodic boundary conditions along the lateral dimensions, the momenta $k_x$ and $k_y$ are good quantum numbers because of the translation symmetry. Therefore, the layer-resolved Chern number can be calculated by projecting the total Chern number into specific layer, which can be written as

$$C_z = \frac{1}{\pi} \sum_{E_m(\mathbf{k}) < E_F < E_n(\mathbf{k})} \int dk_x dk_y \mathrm{Im} \frac{\langle m_\mathbf{k} | \hat{P}_z \partial_{k_x} H_{HSAI} | n_\mathbf{k} \rangle \langle n_\mathbf{k} | \partial_{k_y} H_{HSAI} | m_\mathbf{k} \rangle}{[E_m(\mathbf{k}) - E_n(\mathbf{k})]^2}.$$ 

(4)

Here, $E_F$ is the Fermi energy, $E_{m(n)}(\mathbf{k})$ is the eigenenergy of $H_{HSAI}$ with $|m_\mathbf{k}\rangle$ ($|n_\mathbf{k}\rangle$) the corresponding eigenstates, $\hat{P}_z = |\psi_z\rangle\langle\psi_z|$ is the projecting operator. The integral is performed inside the first Brillouin zone.

Under an electric field $E_z$ along $\hat{z}$-direction, a potential drop occurs inside the HSAI slab along the same direction. The onsite energy in each layer acquires an additional value $eE_z z$ with $z$ the layer index and the total potential drop in the HSAI slab is $\delta U = eE_z L_z$. The orbital magnetization can then be obtained accordingly by using

$$M = \frac{-e}{2\pi h} \sum_{\tilde{E}_m < E_F < \tilde{E}_n} \int dk_x dk_y \mathrm{Im} \frac{(\tilde{E}_m + \tilde{E}_n - 2E_F)}{(\tilde{E}_m - \tilde{E}_n)^2} \langle \tilde{m} | \partial_{k_x} \tilde{H}_{HSAI} | \tilde{n} \rangle \langle \tilde{n} | \partial_{k_y} \tilde{H}_{HSAI} | \tilde{m} \rangle,$$

(5)

where $\tilde{H}_{HSAI} = H_{HSAI} + eE_z z$ with $\tilde{E}_{m(n)}$ and $|\tilde{m}\rangle$ ($|\tilde{n}\rangle$) its eigenenergy and eigenstate, respectively.

Applying a magnetic field $B_z$ along $\hat{z}$-direction introduces a gauge potential to the HSAI lattice and thus breaks the in-plane translation symmetry. Inside each unit cell, HSAI acquires a gauge field $\phi_0 = \int d\mathbf{r} \mathbf{A} \cdot \mathbf{r}/\Psi_0$ with $\Psi_0 = h/(2e)$ the magnetic flux quantum. The total magnetic flux penetrating the HSAI slab is $\phi = B_z L_x L_y$. We adopt the Landau gauge $\mathbf{A} = (-yB_z, 0, 0)$, so the translation symmetry along $\hat{y}$-direction is broken while that along $\hat{x}$-direction sustains. In this case, the charge distribution induced by the magnetic field can be obtained by using the Green's function method, yielding

$$q(z) = \frac{e}{\pi} \sum_{x,y} \int_{-\infty}^{E_F} dE \mathrm{Im} \mathrm{Tr} G^r(E, \mathbf{r}),$$

(6)

where $\mathbf{r} = (x, y, z)$ and the Green's function $G^r(E, \mathbf{r}) = (E + i\eta - H_{HSAI})^{-1}$ with $\eta$ the imaginary line width function. On the other hand, as $k_x$ is still a good quantum number, the charge distribution along $\hat{z}$-direction can be alternatively obtained by using

$$q(z) = \frac{e}{2\pi^2} \sum_{y} \int_{-\infty}^{E_F} dE \int dk_x \mathrm{Im} \mathrm{Tr} G^r(E, k_x, y, z).$$

(7)

Moreover, because only the negative charge originating from electrons are considered here in Eqs. (6) and (7), to derive the unbalanced charge distribution and in turn the polarization, the uniform background charge compensating the positive ions in the lattice has to be removed from the results, which has the form $q_{back} = -\sum_{z=-L_z/2}^{L_z/2} q(z)/L_z$ because of the charge conservation. As a result, the charge distribution has the form $Q(z) = q(z) + q_{back}$. The charge polarization can finally be expressed as $P = \sum_{z=-L_z/2}^{L_z/2} zQ(z)/L_z$.

### Caltulations of the axion term using the hybrid Wannier function

In a HSAI slab, the hybrid Wannier wavefunction $|h_{n,\mathbf{k}}\rangle$ can be constructed from the Bloch wave function. We thus have $|h_{n,\mathbf{k}}\rangle = 1/2\pi \int_{-\pi}^{\pi} dk_z |n_\mathbf{k}\rangle e^{-i(\mathbf{k}\cdot\mathbf{r} + k_z z)}$. In this case, the hybrid Wannier charge center takes the form $z_{n_\mathbf{k}} = \langle h_{n,\mathbf{k}} | z | h_{n,\mathbf{k}} \rangle$[49]. To calculate the non-Abelian Berry curvature, we divide the two-dimensional Brillouin zone into a regular mesh with $b_x$ and $b_y$ being the primitive reciprocal vectors that define the mesh. Then the gauge covariant Berry curvature has the form[55]

$$\tilde{\Omega}_{\mathbf{k}nn}^{xy} = i(\langle \tilde{\partial}_x h_{n,\mathbf{k}} | \tilde{\partial}_y h_{n,\mathbf{k}} \rangle - \mathrm{h.c.}),$$

(8)

where $|\tilde{\partial}_i h_{n,\mathbf{k}}\rangle = (|\tilde{h}_{n,\mathbf{k}+b_i}\rangle - |\tilde{h}_{n,\mathbf{k}-b_i}\rangle)/2$. The wavefunctions constructed by a linear combination of the occupied bands at neighboring mesh point are $|\tilde{h}_{n,\mathbf{k}\pm b_i}\rangle = \sum_{n'}(S_{\mathbf{k},\mathbf{k}\pm b_i}^{nn'})^{-1} \times |h_{n',\mathbf{k}\pm b_i}\rangle$, where the matrix $S_{\mathbf{k},\mathbf{k}'}^{nn'} = \langle h_{n,\mathbf{k}} | h_{n',\mathbf{k}'} \rangle$.

### Green's function method for calculating the differential conductance $G_{ij}$

The differential conductance $G_{ij}$ corresponds to the transmission coefficient $T_{ij}$ from terminal $j$ to terminal $i$, which can be derived by using the non-equilibrium Green's function method. Based on the Landauer-Büttiker formula[48], the transmission coefficient $T_{ij}$ can be expressed as $T_{ij} = \mathrm{Tr}[\Gamma_i G^r \Gamma_j G^a]$, where $\Gamma_{i(j)} = i[\Sigma_{i(j)} - \Sigma_{i(j)}^\dagger]$ is the line width function and $G^r = (G^a)^\dagger = [E_F + i\eta - H_{HSAI} - \sum_i \Sigma_i]^{-1}$ with $E_F$ the Fermi energy, $\eta$ the imaginary line width function and $\Sigma_i$ the self energy due to the coupling to terminal $i$. To incorporate the disorders, we generate random potentials $\delta E \in (-W/2, W/2)$ for the non-magnetic Anderson disorders or $\delta M_z \in (-W_z/2, W_z/2)$ for magnetic Anderson disorders at each site $\mathbf{r}$, then add these random potentials to the Hamiltonian in the Green's functions. The results in the presence of disorders are calculated under 10 times average (Fig. 2f).

## Data availability

The data that support the plots within this paper and other findings of this study are available from the corresponding author upon request. Source data are provided with this paper.

## Code availability

The code that is deemed central to the conclusions is available at https://doi.org/10.24433/CO.7892923.v1.

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

## Acknowledgements

We are grateful for the fruitful discussions with Dr. Zhiqiang Zhang and Prof. Chui-Zhen Chen. H.J. acknowledges the support from the National Key R&D Program of China (Grants No. 2019YFA0308403 and No. 2022YFA1403700) and the National Natural Science Foundation of China (Grant No. 12350401). Y.-H.L acknowledges the support from the Fundamental Research Funds for the Central Universities. X.C.X. acknowledges the support from the Innovation Program for Quantum Science and Technology (Grant No. 2021ZD0302400).

## Author contributions

H.J. conceived the idea of high spin axion insulators after a discussion with Y.-H.L. and X.C.X. S.L. performed calculations with assistance from M.G., Y.-H.L. and H.J. S.L. and Y.-H.L. wrote the manuscript with contributions from all authors. H.J. and X.C.X. supervised the project.

## Competing interests

The authors declare no competing interests.
