## [Peer Review File · Nature Communications]

REVIEWER COMMENTS

Reviewer #1 (Remarks to the Author):

In the manuscript entitled “High spin axion insulator”, the authors proposed a new kind of axion insulators, which appear in high-spin systems extended from the spin-1/2 axion insulator and are protected topologically both by the lattice and time-reversal symmetry, through systematically studying the spin-3/2 and 5/2 systems. Particularly, the authors uncovered several peculiar and important properties, such as a notable axion field to describe the high-spin axion insulators, the absence of gapless quasi-particle excitation, hinting currents, quantum anomaly and others. Moreover, how to design and observe these high-spin axion insulator states is also discussed in the work. From carefully evaluating this work, I think the theoretical results in the present paper are reliable and show high innovations, and are helpful for us to further understand the basic physics of axion insulators and their possible device applications. Thus I support this work to be accepted for publication in Nature Communications.

Reviewer #2 (Remarks to the Author):

The manuscript by Li and coworkers introduces the concept of a High Spin Axion Insulator (HSAI), extending the understanding of axion insulators by proposing a framework within high spin space and highlighting its unique properties. Through theoretical analysis, the study demonstrates that the HSAI retains multiple axion fields protected by combined PT symmetry. This unique axion field is validated through multiple independent methods, including direct calculations employing hybrid Wannier functions, assessments based on layer-specific Chern numbers, and observations of the topological magneto-electric effect. Notably, while the bulk maintains a quantized axion term, the surface exhibits a notably larger axion field alongside an integer Chern number, adjustable through external magnetic field manipulation. Intriguingly, despite possessing an integer surface Chern number, the HSAI lacks guaranteed gapless quasi-particle excitations at its boundaries, suggesting an unconventional response of bulk-boundary correspondence in this spin-3/2 system. Considering the implications and the impact of the manuscript on the community, it has the potential to be published in Nature Communications. However, there are several issues that the authors should clarify.

The manuscript introduces an intriguing extension of axion insulators to the spin-3/2 particle system from an original framework designed for spin-1/2 particles. However, a critical gap lies in the explanation of how the symmetries inherent in the spin-1/2 model are preserved and defined within the extended spin-3/2 model, particularly in the constructed Hamiltonian (Equation 1). It is imperative to rebuild the Hamiltonian from the basis of the spin-3/2 particle system while rigorously

defining the symmetries specific to this system from the outset. Moreover, the significance of the spin-orbit coupling, a crucial parameter influencing axion insulators' origin, is notably absent from the extended model's considerations, which should be explicitly addressed.

The manuscript lacks discussion regarding the axion insulator response dependence concerning the even or odd layers of the systems like MnBi₂Te₄, where only the antiparallel magnetization between external layers induces a topological axion state (Nature Communications 13, 7714 (2022)), a significant omission requiring clarification.

The assertion that the "HSAI state violates the conventional bulk-boundary correspondence" remains unsubstantiated within the manuscript. Figure 2.a displays hinge states seemingly disconnected from the system's symmetry, suggesting a potential restoration of bulk-boundary correspondence rather than its violation. A theoretical demonstration or probe supporting this claim should be included for clarity.

While indirect probes substantiate the existence of the axion insulator phase in the spin-3/2 system, a critical absence is the direct calculation of the Chern-Simons three-form integral (Eq. (1) in reference [44]). It's imperative to address why this integral was not computed and provide a detailed elucidation of the definitions and calculation methods for the "Surface Chern number" and " $\theta^{\text{bulk_CS}}$ " used in the indirect methods to comprehensively validate the findings.

Finally, the methods section raises concerns regarding the calculation of periodic responses (e.g., Bloch waves, Berry curvatures,...) for a non-periodic Hamiltonian (Equation (1)). A comprehensive explanation is warranted to clarify how these calculations are executed for a non-periodic system. Additionally, sharing the code used for these computations on an open-access webpage would greatly benefit the scientific community.

Some minor issues are:

*Clarify PT symmetry's role in protecting axion insulator phases, as per Phys. Rev. B 98, 245117 (2018).

*Rectify discrepancies between k-path representations in figures 3 and their corresponding captions.

List of significant changes

1. To rule out the possibility of the finite size effect, we added a section (Section 3) to the Supplementary Information. We also emphasized this point in the main text on Page 4 (colored blue).
2. We modified our expression at the section “Non-reciprocal conductance” to clarify the relation between the quantized non-reciprocal conductance and the surface Chern number.
3. We added Section 2 to the Supplementary Information discussing the model Hamiltonian construction from symmetry perspective, which is also mentioned in the “Model Hamiltonian for HSAI” section.
4. We added Section 6 to the Supplementary Information calculating the layer-resolved Chern number for even and odd layer HSAI systems, and also pointed this out in the “Model Hamiltonian for HSAI” section.
5. We replotted figure 2a for clarity. The caption corresponding to this figure is also improved. Figure 3 is also replotted.
6. We revised our expression in the second paragraph in the Sec. “Axion term”.
7. We added Refs. 12, 35 and 48 to our reference list in the main text and Ref. 6 in the Supplementary Information.
8. We corrected some typos and errors in both the main text and the Supplementary Information.

All changes are highlighted in revised manuscript.

Reply to Reviewer A

Comment: In the manuscript entitled “High spin axion insulator”, the authors proposed a new kind of axion insulators, which appear in high-spin systems extended from the spin-1/2 axion insulator and are protected topologically both by the lattice and time-reversal symmetry, through systematically studying the spin-3/2 and 5/2 systems. Particularly, the authors uncovered several peculiar and important properties, such as a notable axion field to describe the high-spin axion insulators, the absence of gapless quasi-particle excitation, hinting currents, quantum anomaly and others. Moreover, how to design and observe these high-spin axion insulator states is also discussed in the work. From carefully evaluating this work, I think the theoretical results in the present paper are reliable and show high-level innovations, and are helpful for us to further understand the basic physics of axion insulators and their possible device applications. Thus I support this work to be accepted for publication in Nature Communications. Nevertheless, some suggestions and comments are provided for reference as listed below:

Response: We sincerely appreciate this positive comment. A point-by-point reply to all suggestions and comments is presented below. Corresponding revisions are made in the manuscript, which are colored blue.

Comment: The authors give a description that in Fig. 2b, the topologically protected gapless excitations in lower dimension are completely absent, and think this property results in an unusual quantum anomaly. I think this statement should be clarified further. As described in Fig. 2b, an energy gap in energy spectrum may be induced by the finite-size effect or produced by some special parameters. For example, in the spin-3/2 axion insulator, there is gapless at the Fermi level. These phenomena indicate that the quantum anomaly may be not a common feature in high-spin axion insulators. We suggest the authors address or make clear this issue in the resubmitted version.

Response: Thank you for this insightful comment. Indeed, the energy gap may be induced by the finite-size effect, and the results shown in Fig. S4b for the spin-5/2 axion insulator seems gapless. To rule out this possibility, we first compare the band structure of the spin-3/2 HSAI with that of a Chern insulator by flipping the magnetic moments into parallel case, which supports chiral edge states due to non-zero Chern number $C = 4$. The results for the same size in Figs. R1a and b below show that HSAI in the antiferromagnetic case is gapped while the Chern insulator state in the Ferromagnetic case has gapless edge states inside the band gap. On the other hand, figure R1c shows the band gap for the HSAI as a function of L_y . It is apparent that this band gap is independent of the size L_y , which thus rules out the possibility of the finite-size effect induced gap along y -direction. To further rule out the possibility of the finite-size effect along z -direction, we compare the band gap for the HSAI with the bulk band gap of the Chern insulator after removing the edge bands (red lines in

Fig. R1b). The results plotted in Fig. R1d show that the two band gaps as functions of L_z coincide quantitatively with each other, demonstrating that the band gap in HSAI is also induced by the bulk bands rather than the finite-size effect due to the overlapping between edge states on top and bottom surfaces.

Figure R1. **Size dependence of the energy gap.** **a** and **b**, show the band structures for systems in the antiferromagnetic and ferromagnetic cases, respectively. In the ferromagnetic case, the system is a spin-3/2 Chern insulator with a Chern number $C = 4$. The red lines in **b** denotes the four gapless surface bands. Here, the system size is $L_y = 20$, $L_z = 10$. **c**, energy gap for the HSAI versus L_y . **d**, L_z dependence of the HSAI band gap (red triangle) and of the Chern insulator bulk band gap (blue up-side-down triangle) with $L_y = 20$. The result for the Chern insulator bulk band gap is obtained after removing the surface states (red lines). All parameters for the model Hamiltonian are exactly the same as those in the main text.

We have added this part to the Supplementary Information as Section 3 and also emphasized this in the main text on Page 4.

Comment: The authors describe that the non-reciprocal conductance is a unique characteristics of the high-spin axion insulators. In fact, the non-reciprocal conductance may also appear in other systems having chirality, such as chiral molecules displaying nontrivial features. This indicates the non-reciprocal conductance may be not the unique feature of high-spin axion insulators. Thus how do the authors think that non-reciprocal conductance measurements offer a reliable experimental method to visualize the QHHC in HSAI? Additionally, how to understand the chirality in high-spin axion insulators?

Response: Yes, the non-reciprocal conductance may also appear in many other systems with chirality. There are two crucial differences between the non-reciprocal conductance in high-spin axion insulator and that in other chiral systems. First, the non-reciprocal conductance in high-spin axion insulator is quantized, which is bonded to the surface Chern number $C_{surf}^{top/bot} = \mp(\frac{1}{2} + \frac{3}{2} + \dots + s)$ for a HSAI with spin- s .

Second, the chirality of the non-reciprocal conductance on one side is bonded to its surface Chern number. Because the top and bottom surface Chern numbers are opposite, the values of the quantized non-reciprocal conductance on different surfaces are opposite as shown in Figs. 2a and 2f in the main text. The overall chirality of a high-spin axion insulator is nevertheless not well defined since the system preserve the mirror symmetry $h(\sigma_z)$. Those properties make the quantized non-reciprocal conductance unique to the high-spin axion insulator.

We have modified our expression on Page 4 to make this point clear.

Comment: The authors pointed out that the axion term in high-spin axion insulators mostly originate from localized surface Wannier functions, indicating that the origination of the axion term in low-spin case (i.e., spin-1/2) is much different from the high-spin cases. What is the physical mechanism leading to this difference?

Response: The axion term can in principle be obtained by using the Chern-Simons three form

$$\theta = \frac{1}{4\pi} \int d^3k \epsilon_{ijk} \text{Tr} \left[A_i \partial_j A_k - i \frac{2}{3} A_i A_j A_k \right], \quad (\text{R1})$$

where ϵ_{ijk} is the Levi-Civita symbol, and $A_i = -i\langle u\mathbf{k} | \partial_i | u\mathbf{k} \rangle$ with $|u\mathbf{k}\rangle$ the Bloch wavefunction of occupied bands. This requires that the model Hamiltonian can be solved analytically. Unfortunately, in the high-spin axion insulator, it is extremely challenging to derive the Bloch wavefunction analytically by solving the model Hamiltonian because the dimension of the Hamiltonian is $4s + 2$, which is 8 for spin-3/2 case while 12 for spin-5/2 case (On this point, please also see our response to the fifth comment from reviewer 2). We thus resort to the numerical method developed in *Phys. Rev. B* **101**, 155130 (2020) to calculate the axion term on a slab geometry with periodic boundary condition in the lateral dimensions while open boundary in the perpendicular direction, which replaces the Bloch wavefunction in the upper equation with the Wannier function and recasts above Eq. (R1) into Eq. 3 in the main text. Moreover, the slab geometry is also more realistic in experiment. In this scenario, the total Wannier bands consist of two different categories, those localized inside the surface as denoted by the colorful lines in Figs. 3a and 3f in the main text and those extended into the bulk represented by the black lines in the same figure. The total axion term for a slab can then be separated into two corresponding parts. The value of each part is identical to the Chern number of the corresponding Wannier bands as revealed

in Eq. 3 in the main text. Since the Chern number of the surface Wannier bands in spin-1/2 axion insulator is zero, the surface axion term originated from these Wannier bands is thus zero. By contrast, the Chern number of the surface Wannier bands in the HSAI is non-vanishing, which thus leads to an additional surface axion term different from spin-1/2 axion insulator. The axion terms for different spins and different Wannier bands are summarized in Table. 1 in the main text. Note that the surface Wannier bands only exist in the open boundary case. Conducting a periodic boundary condition along the perpendicular direction by connecting the two surfaces will cancel them and therefore, restore the results calculated by using the Chern-Simons three form.

We have improved our expression on this point in revised manuscript by adding some sentences in the second paragraph in Section “Axion term”.

Comment: The authors described that these high-spin axion insulator states may be detected directly by nano angle-resolved photomission spectroscopy and microscopy. Could the authors propose some real materials apart from the cold-atom systems? Could the antiferromagnetic multiple-layer Van der Waals heterojunctions realize the high-spin axion insulators? The real materials serving as high-spin axion insulators may attract higher attentions from the experimentalists in the related research field.

Response: To realize the high spin axion insulator in real materials, the orbital bands for the model Hamiltonian must come from p , d or even higher orbitals. Furthermore, those orbitals are also required to stay near the Fermi surface without any large energy split. So far, since great efforts have been made in the pursuit of high spin Rarita-Schwinger-Weyl atoms (For example: *National Science Review* 10: nwac121, 2023 and [arXiv:1612.05938v3](https://arxiv.org/abs/1612.05938v3)), this high-spin axion insulator could possibly be realized in Rarita-Schwinger-Weyl atoms if the antiferromagnetic order is introduced. However, the direct realization in real materials needs further first-principle calculations, which is beyond the scope of this work.

Comment: Some minors should be checked and improved in the resubmission. For example: (1) Figure 1d and the associated result are not mentioned in the main text; (2) The formations of the references should be uniformed and some references such as Refs. (24), (26), (29) and others should be corrected; (3) The equation of charge polarization in the last paragraph in Page 5 in the main text should be checked.

Response: Done.

Reply to Reviewer 2

Comment: The manuscript by Li and coworkers introduces the concept of a High Spin Axion Insulator (HSAI), extending the understanding of axion insulators by proposing a framework within high spin space and highlighting its unique properties. Through theoretical analysis, the study demonstrates that the HSAI retains multiple axion fields protected by combined PT symmetry. This unique axion field is validated through multiple independent methods, including direct calculations employing hybrid Wannier functions, assessments based on layer-specific Chern numbers, and observations of the topological magneto-electric effect. Notably, while the bulk maintains a quantized axion term, the surface exhibits a notably larger axion field alongside an integer Chern number, adjustable through external magnetic field manipulation. Intriguingly, despite possessing an integer surface Chern number, the HSAI lacks guaranteed gapless quasi-particle excitations at its boundaries, suggesting an unconventional response of bulk-boundary correspondence in this spin-3/2 system. Considering the implications and the impact of the manuscript on the community, it has the potential to be published in Nature Communications. However, there are several issues that the authors should clarify.

Response: Thank you for carefully reviewing our manuscript and deeming that our work “has potential to be published in Nature Communications”. All issues have been resolved. A point-by-point reply is attached below. Corresponding revisions colored blue are made in the revised manuscript.

Comment: The manuscript introduces an intriguing extension of axion insulators to the spin-3/2 particle system from an original framework designed for spin-1/2 particles. However, a critical gap lies in the explanation of how the symmetries inherent in the spin-1/2 model are preserved and defined within the extended spin-3/2 model, particularly in the constructed Hamiltonian (Equation 1). It is imperative to rebuild the Hamiltonian from the basis of the spin-3/2 particle system while rigorously defining the symmetries specific to this system from the outset. Moreover, the significance of the spin-orbit coupling, a crucial parameter influencing axion insulators' origin, is notably absent from the extended model's considerations, which should be explicitly addressed.

Response: We highly appreciate this comment, which we believe significantly improves our work. In analogy to the topological insulator, spin-orbit coupling is crucial for the realization of the high-spin axion insulator. To construct our model Hamiltonian, we follow the typical strategy of the pursuit of topological insulators. Our model Hamiltonian can be built from the high spin topological insulator preserving both *parity* and *time-reversal* symmetry [*Nat. Phys.* **6**, 284-288(2010)], which has the form

$$H_0 = (m_0 - Bk^2)s_0 \otimes \tau_z + \sum_{i=x,y,z} A_i k_i s_i \otimes \tau_x, \quad (\text{R2})$$

where $k^2 = k_x^2 + k_y^2 + k_z^2$, m_0 , B , $A_{i=x,y,z}$ are system parameters. s_i and τ_i are Pauli matrices acting on spin and orbital spaces. The parity and time-reversal operator for high-spin topological insulator can be defined as $P = \tau_z$ and $T = e^{-is_y}K$ with K the complex conjugate operator. The axion term for this high-spin topological insulator is also $\theta = \pi$, which is protected by the time-reversal symmetry rather than combined lattice and time-reversal symmetry (for this point, please check our reply to comment of the role of PT symmetry). The spin orbital coupling is crucial for this quantized axion field. In the absence of spin-orbital coupling ($A_{x,y,z} = 0$), the upper Hamiltonian in Eq. (R2) is a normal insulator with a vanishing axion term $\theta = 0$. Introducing magnetic ordering into the Hamiltonian explicitly breaks the time-reversal symmetry. However, in the antiferromagnetic case, the combined ***lattice and time-reversal symmetry*** $S = T\tau_{1/2}$ (T is the time-reversal symmetry and $\tau_{1/2}$ is the half translation symmetry along z -direction) is restored, which simultaneously bring an additional exchange term $\Delta\mathbf{m}_s \cdot \mathbf{s} \otimes \tau_0$ to the Hamiltonian, where Δ is the exchange gap between the antiparallel magnetic moment \mathbf{m}_s and the topological electrons with spin \mathbf{s} . The ***minimum*** model Hamiltonian for the high-spin axion insulator combining both topological insulator and antiferromagnetic ordering is thus

$$H = H_0 + \Delta\mathbf{m}_s \cdot \mathbf{s} \otimes \tau_0. \quad (\text{R3})$$

This Hamiltonian constructed from symmetry perspective shares exactly the same form as the spin-1/2 axion insulator.

We have added this part to the Supplementary Information as Section 2 and also mentioned this in the main text. More details can be found there.

Comment: The manuscript lacks discussion regarding the axion insulator response dependence concerning the even or odd layers of the systems like MnBi₂Te₄, where only the antiparallel magnetization between external layers induces a topological axion state (Nature Communications 13, 7714 (2022)), a significant omission requiring clarification.

Response: Thank you for reminding us of this important work. We have cited it properly in revised manuscript. We are also grateful for this comment, which leads us to the interesting ***high Chern number insulator***.

In the odd layer system, the net magnetization is non-vanishing because of the uncompensated magnetic layer, which breaks the combined lattice and time-reversal (S) symmetry. We can simulate this state by using parallel magnetic moments on both top and bottom layers. In order to reveal the difference between them, we explore the layer resolved Chern numbers that can be obtained by using Eq. (4) in the methods. The results displayed in Fig. R2 below show that the layer-resolved Chern numbers for odd

layer systems distribute symmetrically on the top and bottom layers, leading to a vanishing axion field $\theta = 0$ while a nonzero Chern number $C = (s + 1/2)^2$, resembling the odd layer MnBi_2Te_4 . Therefore, the odd layer system is a (high Chern number) Chern insulator instead of an axion insulator. On the contrary, the even layer system is a HSAI with an axion field $\theta = (s + 1/2)^2\pi$ because of the asymmetric layer-resolved Chern numbers.

Figure R2. **Even-odd effect of HSAI.** **a** and **b** are layer-resolved Chern number for spin-3/2 and spin-5/2 HSAI with different layer thicknesses $L_z = 19$ (red triangle) and $L_z = 20$ (blue square). The Chern numbers and axion fields are labeled in the figure.

This part has been added to the Supplementary Information as Section 6 and was also pointed out in the second paragraph on page 3.

Comment: The assertion that the "HSAI state violates the conventional bulk-boundary correspondence" remains unsubstantiated within the manuscript. Figure 2.a displays hinge states seemingly disconnected from the system's symmetry, suggesting a potential restoration of bulk-boundary correspondence rather than its violation. A theoretical demonstration or probe supporting this claim should be included for clarity.

Response: We apologize for the misleading. Figure 2a is a schematic figure that illustrates the possible **current flow** in high-spin axion insulator rather than **electron states**. Owing to the non-zero Berry curvature, the electron on the side surface experience a Goos-Hänchen shift at the hinge and thus propagates oppositely [*National Science Review* 10, nwad025 (2023)], resulting in a quantized helical hinge current identical to the surface Chern number $C_{surf} = \pm(1/2 + 3/2 + \dots + s)$. Nevertheless, this hinge current is not carried by a low-dimension edge state because the energy band is fully gapped as shown in Fig. 2b in the main text for the spin-3/2 high spin axion insulator though its surface Chern number is $C_{surf} = \pm 2$, which violates the conventional bulk-boundary correspondence that an integer Chern number must

guarantee a gapless quasi-particle excitation in lower dimension. For this point, please also check our response to the first comment from reviewer 1.

To avoid any misleading, we revised the schematic figure in Fig. 2a in the main text and clarified this point accordingly in the caption.

Comment: While indirect probes substantiate the existence of the axion insulator phase in the spin-3/2 system, a critical absence is the direct calculation of the Chern-Simons three-form integral (Eq. (1) in reference [44]). It's imperative to address why this integral was not computed and provide a detailed elucidation of the definitions and calculation methods for the “Surface Chern number” and “theta term^{bulk_CS}” used in the indirect methods to comprehensively validate the findings.

Response: Indeed, the axion term can in principle be obtained by using the Chern-Simons three-form

$$\theta = \frac{1}{4\pi} \int d^3k \epsilon_{ijk} \text{Tr} \left[A_i \partial_j A_k - i \frac{2}{3} A_i A_j A_k \right], \quad (\text{R4})$$

Where ϵ_{ijk} is the Levi-Civita symbol, and $A_i = -i \langle u\mathbf{k} | \partial_i | u\mathbf{k} \rangle$ with $|u\mathbf{k}\rangle$ the Bloch wavefunction of occupied bands. To do so, the system needs to be periodic along all three directions and the model Hamiltonian has to be solved analytically to avoid the random phase problem in the Bloch wavefunction $|u\mathbf{k}\rangle$. In the spin-1/2 axion insulator, one can solve the Hamiltonian and derive the analytical Bloch wavefunction very straightforwardly since the Hamiltonian is 4×4 . Unfortunately, because the dimension of the Hamiltonian for the high-spin axion insulator equals to $2 * (2s + 1)$ (the factor 2 originates from the orbital space), in the high-spin case the Hamiltonian is at least 8×8 (when $s = 3/2$), or even larger. It is thus extremely challenging to calculate the Bloch wavefunction analytically. On the other hand, the experiment in axion insulator is typically conducted on a few-layer system, which is actually a slab in reality. We therefore explore a slab geometry that agrees perfectly with experiments to calculate the axion term numerically by using the method developed in *Phys. Rev. B* **101**, 155130 (2020). Since the Wannier function can be constructed by using the Bloch wavefunctions, the upper Chern-Simons three form in Eq. (R4) can be rewritten into

$$\theta_{CS}^{slab} = -\frac{1}{L_z} \int d^2\mathbf{k} \sum_n [z_{n\mathbf{k}} \tilde{\Omega}_{knn}^{xy}], \quad (\text{R5})$$

where $z_{n\mathbf{k}}$ is the hybrid Wannier charge center along the perpendicular direction and $\tilde{\Omega}_{knn}^{xy}$ is corresponding non-Abelian Berry curvature. However, as indicated by the colorful lines in Figs. 3a and 3f, open boundary condition along the perpendicular direction introduces some additional Wannier bands that are well localized on the top and bottom surfaces. Those surface Wannier bands do not exist in periodic boundary

case, and will disappear if we restore the periodic boundary by connecting the top and bottom surfaces. Consequently, we can separate those Wannier functions as well as the axion term into two different parts: one coming from the surface Wannier bands denotes as θ_{CS}^{surf} while the other from the bulk Wannier bands denotes as θ_{CS}^{bulk} . Moreover, the value of each part is determined by the Chern number of the corresponding Wannier bands, which leads to a total axion term $\theta_{CS}^{slab} = (s + 1/2)^2\pi$. Note that θ_{CS}^{bulk} is identical to the axion term calculated directly by using the Chern-Simons three form in the infinite-size limit and the total axion term for the slab geometry in experiment is $\theta_{CS}^{slab} = \theta_{CS}^{surf} + \theta_{CS}^{bulk}$.

To clarify this point, we improved our expression by adding a few sentences to the section of “Axion term” in the main text.

Comment: Finally, the methods section raises concerns regarding the calculation of periodic responses (e.g., Bloch waves, Berry curvatures,...) for a non-periodic Hamiltonian (Equation (1)). A comprehensive explanation is warranted to clarify how these calculations are executed for a non-periodic system. Additionally, sharing the code used for these computations on an open-access webpage would greatly benefit the scientific community.

Response: All quantities except the charge polarization $Q(z)$ in the methods are calculated on a slab geometry with periodic boundary in the lateral dimensions (integral over k_x and k_y) while open boundary condition in the perpendicular direction. Because an external magnetic field breaks the in-plane translation symmetry, the charge polarization $Q(z)$ is calculated by incurring open boundary condition along the perpendicular direction and y -direction since we used the Landau gauge while maintain the periodic boundary condition along x -direction.

We have uploaded our codes that can reproduce all figures in our manuscript to Github (link: <https://github.com/Asntz/High-spin-axion-insulator>).

Comment: Some minor issues are:

*Clarify PT symmetry's role in protecting axion insulator phases, as per Phys. Rev. B 98, 245117 (2018).

Response: We apologize for the misleading. The quantized axion field is protected by the combined lattice and time-reversal symmetry $S = T\tau_{1/2}$ with $\tau_{1/2}$ the half translation symmetry along z -direction. As shown in *Journal of Applied Physics* 129, 141101 (2021), the Lagrangian for an insulator can be written as

$$L = \int d^4\mathbf{r} \frac{1}{8\pi} \left(\epsilon E^2 - \frac{B^2}{\mu} \right) + \frac{\theta}{2\pi} \frac{\alpha}{2\pi} \mathbf{E} \cdot \mathbf{B}, \quad (\text{R6})$$

where E and B are conventional electro-magnetic field, ϵ and μ are permittivity and permeability, α is the fine structure constant, and θ is the axion field. The axion field can be expressed into a Chern-Simons three form shown in Eqs. (R1) and (R4). It is 2π periodic under a gauge transformation (under periodic boundary condition but not open boundary case). Shifting the axion field by 2π does not alter the physical quantity denoted by this Lagrangian. It is obvious that the $\mathbf{E} \cdot \mathbf{B}$ flips sign under the operation of $S = T\tau_{1/2}$ symmetry, in analogy to the case under time-reversal operation. Consequently, if the system preserves this S symmetry, the axion field θ has also only two different values: $\theta = 0$ for normal insulators and $\theta = \pi$ for axion insulators, which becomes $\theta = -\pi$ (equivalent to $\theta = \pi$) under the S operation. The role of the S symmetry in protecting the axion insulator phases can be found in many works such as *Phys. Rev. Lett.* **122**, 206401(2019) and *Phys. Rev. B* **98**, 245117(2018), which we have cited properly in revised manuscript.

In our model, we consider a slab with open boundary condition along z -direction instead of a three-dimensional system. The $S = T\tau_{1/2}$ symmetry is broken in this slab geometry while the system preserves the combined parity and time-reversal (PT) symmetry. This PT symmetry does not ensure a quantized axion field since the $\mathbf{E} \cdot \mathbf{B}$ term remains unchanged under this operation. Consequently, the axion term calculated on a slab deviates from the quantized value as shown in Figs. 3b, e and g in the main text. However, in the infinite-size limit, the $S = T\tau_{1/2}$ symmetry is restored and thus the axion field (as well as the coefficient of the topological magnetoelectric response) quantizes to $\left(s + \frac{1}{2}\right)^2 \pi$ in the infinite size limit.

We corrected our expression of the roles of PT symmetry and S symmetry in the main text.

*Rectify discrepancies between k-path representations in figures 3 and their corresponding captions.

Response: Done.

REVIEWERS' COMMENTS

Reviewer #1 (Remarks to the Author):

Since the authors have already improved their work largely according to all Reviewers' comments and addressed all comments reasonably and successfully, I happily support its publication in Nat. Commun. as it is.

Reviewer #2 (Remarks to the Author):

I have reviewed the response provided by the authors to the reviewer comments on the manuscript introducing the concept of a High Spin Axion Insulator. I am pleased to note that the authors have addressed the weak points identified in the initial review and have taken into account all the suggestions for improvement.

The revised response now includes a more detailed explanation of how the symmetries specific to the spin-3/2 model are defined and preserved within the extended Hamiltonian. In addition, the authors have provided a thorough discussion on the role of SOC in the HSAI model and the calculation of the Chern-Simons integral, enhancing the understanding of the research findings.

Considering the enhancements made in the revised version of the manuscript, I am of the opinion that it is now suitably poised for publication in Nature Communications.

Reviewer #2 (Remarks on code availability):

The authors have uploaded their codes that can reproduce all figures in our manuscript to Github link:

<https://github.com/Asntz/High-spin-axion-insulator>